# Reversibility of quantum resources through probabilistic protocols

**Bartosz Regula** [1] ✉ **& Ludovico Lami** [2,3,4] ✉

Among the most fundamental questions in the manipulation of quantum resources such as entanglement is the possibility of reversibly transforming all resource states. The key consequence of this would be the identification of a unique entropic resource measure that exactly quantifies the limits of achievable transformation rates. Remarkably, previous results claimed that such asymptotic reversibility holds true in very general settings; however, recently those findings have been found to be incomplete, casting doubt on the conjecture. Here we show that it is indeed possible to reversibly inter-convert all states in general quantum resource theories, as long as one allows protocols that may only succeed probabilistically. Although such transformations have some chance of failure, we show that their success probability can be ensured to be bounded away from zero, even in the asymptotic limit of infinitely many manipulated copies. As in previously conjectured approaches, the achievability here is realised through operations that are asymptotically resource non-generating, and we show that this choice is optimal: smaller sets of transformations cannot lead to reversibility. Our methods are based on connecting the transformation rates under probabilistic protocols with strong converse rates for deterministic transformations, which we strengthen into an exact equivalence in the case of entanglement distillation.

How to measure and compare quantum resources? As evidenced by the plethora of commonly used quantifiers of resources such as entanglement[1,2], this seemingly basic question has many possible answers, and it may appear as though there is no unambiguous way to resolve it. However, it is important to keep in mind that, besides simply assigning some numerical value to a given quantum state, one often wishes to compare quantum resources operationally. Is it then possible for a resource quantifier to indicate exactly how difficult it is to convert one resource state into another? Following this pathway is reminiscent of the operational approach used to study thermo-dynamics, where indeed a unique resource measure − the entropy − emerges naturally from basic axioms[3,4]. A phenomenon that is inti-mately connected with the existence of such a measure is reversibility:

two comparable states of equal entropy can always be connected by a reversible adiabatic transformation[3,4].

Reversibility was observed also in the asymptotic manipulation of quantum entanglement of pure states[5], prompting several conjectures about the connections between entanglement and thermodynamics, and in particular about the existence of a unique operational measure of entanglement that would mirror the role of entropy[6–10]. In the asymptotic setting, reversibility is typically understood in terms of asymptotic transformation rates $r(\rho \to \omega)$: given many copies of a state $\rho$, how many copies of another state $\omega$ can we obtain per each copy of $\rho$? The question of resource reversibility then asks whether $r(\rho \to \omega) = r(\omega \to \rho)^{-1}$, meaning that exactly as many copies of $\omega$ can be obtained in the transformation as are needed to transform them back

[1]Mathematical Quantum Information RIKEN Hakubi Research Team, RIKEN Cluster for Pioneering Research (CPR) and RIKEN Center for Quantum Computing (RQC), Wako, Saitama 351-0198, Japan. [2]QuSoft, Science Park 123, Amsterdam 1098 XG, The Netherlands. [3]Korteweg–de Vries Institute for Mathematics, University of Amsterdam, Science Park 105-107, Amsterdam 1098 XG, The Netherlands. [4]Institute for Theoretical Physics, University of Amsterdam, Science Park 904, Amsterdam 1098 XH, The Netherlands. ✉e-mail: bartosz.regula@gmail.com; ludovico.lami@gmail.com

into $\rho$. Although entanglement of noisy states may exhibit irreversibility in many contexts[11–14], hopes persisted that an operational approach allowing for universal reversibility could be constructed, leading to the identification of a unique asymptotic measure of entanglement[15].

A remarkable axiomatic framework emerged, first for entanglement[9,10] and later for more general quantum resources[16], which claimed that reversibility can indeed always be achieved under suitable assumptions. Such a striking property would not only establish a unique entropic measure of quantum resources, but also connect the broad variety of different resources in a common operational formalism. However, issues have transpired in parts of the proof of these results[17,18], putting this general reversibility into question. As of now, there is no known framework that can establish the reversibility of general quantum resource theories — and in particular quantum entanglement — even under weaker assumptions. What is more, recent results demonstrated an exceptionally strong type of irreversibility of entanglement[14], casting doubt on the very possibility of recovering reversible manipulation whatsoever.

In this work, we resolve the question by constructing the first complete reversible framework for general quantum resources, including entanglement. Our setting closely resembles the original assumptions of the reversibility conjectures[9,10,16], with only one change: we allow probabilistic conversion protocols. That is, we study transformations which allow for some probability of failure, and we demonstrate that in this setting the conversion rates are exactly given by the entropic resource measure known as the regularised relative entropy, identifying it as the unique operational resource quantifier.

The use of probabilistic protocols is a distinguishing feature of our approach, necessitating a careful consideration of how exactly to quantify the asymptotic rates of such transformations. We employ a definition which focuses on the number of copies of states that are undergoing the transformation, discounting the success probability of the protocols. Although seemingly more permissive than some previously used definitions, we explicitly ensure that the protocols that we study are not unphysically difficult to realise: we only allow transformations whose probability of failure does not become prohibitively large, in the sense that there always remains a constant non-zero chance of successful conversion, even when manipulating an unbounded number of quantum states. Such rates are well behaved and closely connected to conventional asymptotic transformation rates studied in quantum information.

We stress that, although conceptually similar, our approach follows an alternative pathway that does not exactly recover the reversibility conjectured in refs. 9,10,16, as the latter relied only on strictly deterministic transformation rates. However, in light of the similarities and relations that we establish between probabilistic and deterministic rates, we consider our results to be strong supporting evidence in favour of reversibility being an achievable phenomenon in general quantum resource theories.

On the technical side, the way we avoid issues associated with the generalised quantum Stein's lemma[19] that undermined the original reversibility claims, is to use only the strong converse part of the lemma, which is still valid[17]. Strong converse rates are typically understood as general no-go limitations on resource transformations, but here we turn them into achievable rates precisely by employing probabilistic protocols. For the special case of entanglement distillation, we show that these two concepts — strong converse rates in deterministic transformations on one side, and probabilistic conversion rates on the other — are exactly equivalent, which holds true also in the most practically relevant settings of entanglement manipulation such as under local operations and classical communication (LOCC).

## Results

### Resource transformation rates

Quantum resource theories represent various settings of restricted quantum information processing[2]. Let us denote by $\mathbb{O}$ the set of operations which are freely allowed within the physical setting of the given resource theory. Our discussion of reversibility will require a specific choice of $\mathbb{O}$, but for now it may be understood as a general set of permitted operations.

The deterministic transformation rate $r_{p=1}(\rho \to \omega)$ is defined as the supremum of real numbers $r$ such that $n$ copies of the state $\rho$ can be converted into $\lfloor rn \rfloor$ copies of the target state $\omega$ using the free operations $\mathbb{O}$. The conversion here is assumed to be deterministic, i.e. all transformations are realised by completely positive and trace-preserving maps (quantum channels). However, the process is only required to be approximate, in the sense that some error $\varepsilon_n$ is allowed in the transformation, as long as it vanishes in the limit as $n \to \infty$.

In many practical contexts, one may be willing to relax the assumption that the error must tend to zero — it may, for instance, be appealing to tolerate some manageable error in the transformation if it could lead to increased transformation rates. The ultimate upper bound that constrains the improvements that can be gained through such trade-offs is represented by the strong converse rate $r^\dagger_{p=1}(\rho \to \omega)$. It is defined as the least rate $r$ such that, if we attempt the conversion $\rho^{\otimes n} \to \omega^{\otimes \lfloor r'n \rfloor}$ at any larger rate $r' > r$, then even approximate transformations with large error become impossible.

Another common way to increase the capabilities in resource manipulation is to allow for probabilistic transformations[20–26]. Probabilistic protocols in quantum information theory are represented by a collection of completely positive, but not necessarily trace-preserving maps $\{\mathcal{E}^{(i)}\}_i$, such that the total transformation $\sum_i \mathcal{E}^{(i)}$ preserves the trace. We say that $\rho$ can be converted to $\omega$ if there exists a free probabilistic operation $\mathcal{E}^{(i)} \in \mathbb{O}$ such that $\frac{\mathcal{E}^{(i)}(\rho)}{\mathrm{Tr}\,\mathcal{E}^{(i)}(\rho)} = \omega$; the probability of this transformation is $p = \mathrm{Tr}\,\mathcal{E}^{(i)}(\rho)$. The question then is how to exactly define the asymptotic rate of such protocols.

Consider a sequence of probabilistic operations $(\mathcal{E}_n)_n$ such that each $\mathcal{E}_n \in \mathbb{O}$ converts $\rho^{\otimes n}$ to a state which is $\varepsilon_n$-close to the target state $\omega^{\otimes \lfloor rn \rfloor}$ with the error vanishing asymptotically. We will write $p_n = \mathrm{Tr}\,\mathcal{E}_n(\rho^{\otimes n})$ for the probability of successful conversion. One way to quantify the rate of such a protocol is to count the average number of copies of quantum states needed to realise the transformation, which means that our rate would be given by $p_n r$ rather than just $r$, *since the protocol needs to be repeated $1/p_n$ times on average to ensure success*. But one may argue that there is an issue with such a definition: is it fair to say that manipulating $n$ copies of a quantum state about $1/p_n$ times is as difficult as manipulating the larger number of $n/p_n$ copies all at once? This definition of a rate would make it seem so, since it counts the total number of copies needed in the protocol, and disregards the question of how many of those copies need to be coherently manipulated together. If the rate is supposed to quantify the difficulty in performing a state transformation, then this may not be an accurate assessment, considering that it is the manipulation of quantum states, rather than their generation, that is typically the bottleneck in practical quantum information processing.

An alternative way is then to simply say that, if $\rho^{\otimes n}$ is approximately converted to $\omega^{\otimes \lfloor rn \rfloor}$ — even probabilistically — then the rate is $r$. This definition focuses on the number of copies of states that are being transformed at once, and it does not count the probability $p_n = \mathrm{Tr}\,\mathcal{E}_n(\rho^{\otimes n})$ itself as part of the rate. However, there is again a potential issue with this approach, as leaving the probability of the transformation unconstrained effectively allows for conditioning on exponentially unlikely events, which then makes possible transformations that are conventionally known to be unachievable[26]. Such a phenomenon happens when the overall probability of success $p_n$ becomes vanishingly small. In order to

exclude such unphysical protocols which cannot be implemented in practice, it thus becomes necessary to carefully constrain the success probability.

Our approach will then aim to find a middle ground: we will quantify rates in a way that only counts the number of transformed states, but we will explicitly forbid the possibility of the success probability becoming unphysically small. Specifically, to ensure that any considered protocol remains practically realisable, we will assume that the conversion probability is bounded away from zero. We thus consider probabilistic transformation rates with non-vanishing probability, $r_{p>0}(\rho \to \omega)$, defined as

$$
\begin{aligned}
r_{p>0}(\rho \to \omega) := \sup_{(\mathcal{E}_n)_n} \Bigg\{ r \,\Bigg|\, & \mathcal{E}_n \in \mathbb{O}, \\
& \lim_{n\to\infty} F\left( \frac{\mathcal{E}_n(\rho^{\otimes n})}{\operatorname{Tr} \mathcal{E}_n(\rho^{\otimes n})}, \omega^{\otimes \lfloor rn \rfloor} \right) = 1, \\
& \liminf_{n\to\infty} \operatorname{Tr} \mathcal{E}_n(\rho^{\otimes n}) > 0 \Bigg\},
\end{aligned}
\tag{1}
$$

where $F$ denotes fidelity. We stress that this imposes a strong restriction on the allowed protocols, as the overall probability of success must remain larger than a positive constant, even in the asymptotic limit where the number of transformed copies grows to infinity. In short, the number of copies of $\rho$ that need to be manipulated at once to obtain $m$ copies of $\omega$ is asymptotically $n = m/r_{p>0}(\rho \to \omega)$, while the total number of copies of $\omega$ that need to be generated is that number times a constant overhead.

To further motivate our choice of definition of a probabilistic rate, let us compare it with the two deterministic rates introduced in this section. The fact that the deterministic transformation rate $r_{p=1}$ is the smallest of the three types is clear from the definition. However, there is no obvious relation between the strong converse rate and the probabilistic one. We can nevertheless show that the rates actually form a hierarchy:

$$
r_{p=1}(\rho \to \omega) \le r_{p>0}(\rho \to \omega) \le r_{p=1}^{\dagger}(\rho \to \omega).
\tag{2}
$$

This demonstrates in particular that the probabilistic rate $r_{p>0}$ is well behaved, as it does not exceed conventional limitations imposed by strong converse rates. It also naturally fits into the information-theoretic framework for asymptotic transformations and may even provide a tighter restriction on deterministic transformation rates than those coming from strong converse bounds.

## Free operations and reversibility

The asymptotic transformation rates depend heavily on the choice of the free operations $\mathbb{O}$. Typically, practically relevant choices of free operations are subsets of resource–non-generating (RNG) operations, defined as those maps $\mathcal{E}$ (possibly probabilistic ones) that satisfy $\frac{\mathcal{E}(\sigma)}{\operatorname{Tr}\mathcal{E}(\sigma)} \in \mathbb{F}$ for all $\sigma \in \mathbb{F}$. Here, $\mathbb{F}$ stands for the set of free (resourceless) states of the given theory. The definition of RNG operations then means that these maps are not allowed to generate any resources for free, which is a very basic and undemanding assumption to make.

The framework of refs. 9,10,16 studied the manipulation of quantum resources under transformations which slightly relax the above constraint, imposing instead that small amounts of resources may be generated, as long as they vanish asymptotically. Specifically, let us consider the resource measure known as generalised (global) robustness $R_{\mathbb{F}}^{g}$, defined as[27]

$$
R_{\mathbb{F}}^{g}(\rho) := \inf\left\{ \lambda \in \mathbb{R}_{+} \,\middle|\, \frac{\rho + \lambda\omega}{1+\lambda} \in \mathbb{F},\ \omega \in \mathbb{D} \right\},
\tag{3}
$$

where $\mathbb{D}$ denotes the set of all states. The $\delta$-approximately resource–non-generating operations $\mathbb{O}_{\mathrm{RNG}_{\delta}}$ are then all maps $\mathcal{E}$ such that

$$
R_{\mathbb{F}}^{g}\left( \frac{\mathcal{E}(\sigma)}{\operatorname{Tr}\mathcal{E}(\sigma)} \right) \le \delta \quad \forall \sigma \in \mathbb{F}.
\tag{4}
$$

Finally, the transformation rates under asymptotically resource–non-generating maps $\mathbb{O}_{\mathrm{ARNG}}$, whether deterministic or probabilistic, are defined as those where each transformation $\rho^{\otimes n} \to \omega^{\otimes \lfloor rn \rfloor}$ is realised by a $\delta_n$-approximately RNG operation, with $\delta_n \to 0$ in the limit as $n \to \infty$. We will denote deterministic rates under such operations as $r_{p=1}(\rho \xrightarrow[\mathrm{ARNG}]{} \omega)$, and analogously for the probabilistic rates $r_{p>0}$.

The main reason to study asymptotically RNG operations is their conjectured reversibility[9,10,16]. Specifically, the claim is that the deterministic rates always equal

$$
r_{p=1}\left( \rho \xrightarrow[\mathrm{ARNG}]{} \omega \right) \overset{?}{=} \frac{D_{\mathbb{F}}^{\infty}(\rho)}{D_{\mathbb{F}}^{\infty}(\omega)},
\tag{5}
$$

where $D_{\mathbb{F}}^{\infty}$ denotes the regularised relative entropy of a resource,

$$
D_{\mathbb{F}}^{\infty}(\rho) := \lim_{n\to\infty} \frac{1}{n} \inf_{\sigma \in \mathbb{F}} D(\rho^{\otimes n} \,\|\, \sigma_n)
\tag{6}
$$

with $D(\rho \,\|\, \sigma) = \operatorname{Tr}\rho(\log\rho - \log\sigma)$ being the quantum relative entropy. This would precisely identify $D_{\mathbb{F}}^{\infty}$ as the unique resource measure in the asymptotic setting. However, this conjecture relied crucially on the generalised quantum Stein's lemma[19], in whose proof a gap was recently discovered[17]. Hence, the statement in Eq. (5) is not known to be true[17].

One may also wonder whether there are other possible candidates for operations that could lead to reversibility. This is especially relevant since the asymptotically resource non-generating maps $\mathbb{O}_{\mathrm{ARNG}}$ are defined in an axiomatic way, and it may be appealing to study smaller classes of transformations constructed through more practically-minded considerations. However, such a possibility has been ruled out: in the context of deterministic transformations, essentially all sets of operations smaller than $\mathbb{O}_{\mathrm{ARNG}}$ have been shown to lead to an irreversible theory of entanglement[14]. Importantly, the choice of the measure $R_{\mathbb{F}}^{g}$ in the definition of $\mathbb{O}_{\mathrm{ARNG}}$ is crucial, and even a small change of the resource quantifier can preclude reversibility. What this means is that any reversible theory of entanglement must actually generate exponentially large amounts of entanglement according to certain measures[14]. But even such entanglement generation is not enough on its own: if one requires the considered transformations to also be 'dually' resource non-generating (i.e., in the Heisenberg picture), then reversibility is impossible, even if one permits generating entanglement akin to $\mathbb{O}_{\mathrm{ARNG}}$[28]. On the other hand, choosing more permissive types of operations, such as ones where the generated resources are quantified with the relative entropy $D_{\mathbb{F}}^{\infty}$, may be too lax of a constraint, as such a theory trivialises by allowing for the distillation of unbounded amounts of entanglement[10]. Altogether, this provides a strong motivation to study reversibility precisely under the class $\mathbb{O}_{\mathrm{ARNG}}$, as it constitutes a 'Goldilocks' set of operations that may allow for reversible manipulation while maintaining reasonable restrictions on the allowed transformations.

## Probabilistic reversibility

The conjectured resource reversibility (Eq. (5)) was formulated in a remarkably general manner. The original claim was meant to apply not only to entanglement, but also to more general quantum resources, as long as the set $\mathbb{F}$ satisfies a number of mild assumptions – notably, it must be convex, and it must be such that the tensor product of any two free states remains free, as does their partial trace[16,19]. These are weak

assumptions obeyed by the vast majority of theories of practical interest.

Our main result is a general probabilistic reversibility of quantum resources under the exact same assumptions.

**Theorem 1.** For all quantum states $\rho$ and $\omega$, the transformation rate with non-vanishing probability of success under asymptotically resource–non-generating operations satisfies

$$r_{p > 0}(\rho \xrightarrow{\mathrm{ARNG}} \omega) = \frac{D_{\mathbb{F}}^{\infty}(\rho)}{D_{\mathbb{F}}^{\infty}(\omega)}. \tag{7}$$

This implies in particular a general reversibility of state transformations: $r_{p > 0}(\rho \xrightarrow{\mathrm{ARNG}} \omega) = r_{p > 0}(\rho \xrightarrow{\mathrm{ARNG}} \omega)^{-1}$ for all pairs of states.

Both the converse and the achievability parts of this result make use of the asymptotic equipartition property for the generalised robustness, which was shown by Brandão and Plenio (Proposition II.1 in ref. [19]) and independently by Datta (Theorem 1 in ref. [29]). This property says that, under a suitable 'smoothing', the generalised robustness $R_{\mathbb{F}}^{g}$ converges asymptotically to the regularised relative entropy of the resource:

$$\lim_{\varepsilon \to 0} \limsup_{n \to \infty} \frac{1}{n} \min_{\frac{1}{2}\|\omega_n - \omega^{\otimes \lfloor rn \rfloor}\|_1 \leq \varepsilon} \log\left(1 + R_{\mathbb{F}}^{g}(\omega_n)\right) = r\, D_{\mathbb{F}}^{\infty}(\omega). \tag{8}$$

Importantly, this finding directly leads to the strong converse of the generalised quantum Stein's lemma (Corollary III.3 in ref. [19]), but it does not appear to be enough to deduce the main achievability part of the lemma[17], which underlies the previous reversibility conjectures. Our main contribution here is to show that the strong converse part is sufficient to show the reversibility of quantum resources, as long as probabilistic protocols are allowed.

**Proof sketch of Theorem 1.** The converse direction relies on the strong monotonicity properties of the generalised robustness $R_{\mathbb{F}}^{g}$ as well as the aforementioned asymptotic equipartition property (8). This follows a related approach that was recently used to study post-selected probabilistic transformation rates[26], and here we extend it to asymptotically resource–non-generating transformations ARNG. A point of note is that standard techniques for upper bounding transformations rates, based on the asymptotic continuity of the relative entropy[1,30,31], do not seem to be sufficient to establish a converse bound on probabilistic rates (see Appendix H in ref. [26]). Our approach requires the use of a different toolset that explicitly makes use of the features of $R_{\mathbb{F}}^{g}$.

For the achievability part of the theorem, we use the exact calculation of the strong converse exponent in the generalised quantum Stein's lemma[19]. The lemma is concerned with the distinguishability of many copies of a quantum state $\rho^{\otimes n}$ against all states in the set of the free states $\mathbb{F}$. The result of ref. [19] then says that, for every resource theory, there exists a sequence of measurement operators $(A_n)_n$ such that $\mathrm{Tr}(A_n \rho^{\otimes n}) \geq 1 - \delta_n$ and

$$-\frac{1}{n} \log \sup_{\sigma \in \mathbb{F}} \mathrm{Tr}(A_n \sigma) \xrightarrow{n \to \infty} D_{\mathbb{F}}^{\infty}(\rho). \tag{9}$$

Here, $\delta_n$ denotes the probability of incorrectly guessing that $\rho^{\otimes n}$ is a free state, while the quantity in Eq. (9) characterises the opposite error of incorrectly guessing that a free state is $\rho^{\otimes n}$. The issue here is that the proof of ref. [19], and hence also Eq. (9), is only valid in the strong converse regime: this means that the error $\delta_n$ is not guaranteed to vanish, but it may actually tend to a constant arbitrarily close to 1. This prevents a direct application of previous deterministic results[16].

What we do instead is define probabilistic operations of the form

$$\mathcal{E}_n(\tau) := \mathrm{Tr}(A_n \tau)\, \omega_n + \mu_n\, \mathrm{Tr}[(\mathbb{1} - A_n)\tau]\, \pi_n, \tag{10}$$

where: $\omega_n$ are states appearing in (8) which are $\varepsilon_n$-close to the target states $\omega^{\otimes \lfloor nD_{\mathbb{F}}^{\infty}(\rho)/D_{\mathbb{F}}^{\infty}(\omega) \rfloor}$, $\pi_n$ are some suitably chosen states, and $\mu_n \in [0, 1]$ are parameters to be fixed.

The basic idea is then that by decreasing $\mu_n$, we can make the output of this operation closer to $\omega_n$, even when $\mathrm{Tr}(A_n \rho^{\otimes n}) \nrightarrow 1$. However, one cannot just decrease $\mu_n$ arbitrarily, as the maps $\mathcal{E}_n$ must be ensured to be free operations. Our crucial finding is that $\mu_n$ can always be chosen so that $\mu_n \xrightarrow{n \to \infty} 0$ while the operations $\mathcal{E}_n$ generate asymptotically vanishing amounts of resources and the overall probability of success does not vanish. This means precisely that the sequence $(\mathcal{E}_n)_n$ is an ARNG protocol that realises the desired conversion.

The complete proof of Theorem 1 can be found in Section II of the Supplementary Information.

## Optimality of ARNG transformations

To provide a stronger motivation for the choice of asymptotically resource–non-generating operations in the study of reversible transformations, we can show that this set of operations is essentially the smallest possible: more restrictive types of operations cannot lead to reversibility in general, even under probabilistic transformations.

One natural way to constrain the allowed resource transformations is to forbid resource generation − that is, instead of asymptotically resource–non-generating maps, consider strictly resource-non-generating ones. An even more fine-grained restriction can be obtained by allowing for approximate resource generation, but choosing a more restrictive resource measure with which to quantify the generated resources. To be precise, let us consider a modified notion of $\delta$-approximately RNG transformations that we will call $\mathbb{O}_{\mathrm{RNG},\delta,s}$. They are defined in exactly the same manner as ARNG rates, but instead of constraining the generalised robustness $R_{\mathbb{F}}^{g}$ as in Eq. (4), we impose that

$$R_{\mathbb{F}}^{s}\left(\frac{\mathcal{E}(\sigma)}{\mathrm{Tr}\,\mathcal{E}(\sigma)}\right) \leq \delta \quad \forall \sigma \in \mathbb{F}, \tag{11}$$

where $R_{\mathbb{F}}^{s}$ denotes the standard robustness[27]

$$R_{\mathbb{F}}^{s}(\rho) := \inf\left\{\lambda \in \mathbb{R}_+ \,\middle|\, \frac{\rho + \lambda\sigma}{1 + \lambda} \in \mathbb{F}, \sigma \in \mathbb{F}\right\}. \tag{12}$$

This measure is very similar to the generalised robustness of Eq. (3), but the state $\sigma$ is now required to be free; because of this, it holds that $R_{\mathbb{F}}^{s}(\rho) \geq R_{\mathbb{F}}^{g}(\rho)$ in general, making the constraint in (11) potentially more restrictive than before. We stress that strictly resource–non-generating transformations $\mathbb{O}_{\mathrm{RNG}}$ are a subset of $\mathbb{O}_{\mathrm{RNG},\delta,s}$, so all irreversibility results shown for the latter apply also to the former.

We can use this to define modified transformation rates $r\left(\rho \xrightarrow{\mathrm{ARNG,s}} \omega\right)$ as those realised under $\mathbb{O}_{\mathrm{RNG},\delta_n,s}$ transformations with $\delta_n \to 0$ as $n \to \infty$.

By extending the methods that we used previously in the study of deterministic irreversibility[14], we can show the following.

**Theorem 2.** Even in the probabilistic setting, general reversibility is not possible under operations that do not generate any resources or ones that only generate asymptotically vanishing amounts of resources according to the standard robustness. Specifically, in the resource

theory of entanglement there exist states $\rho, \omega$ such that

$$r_{p>0}\left(\rho \xrightarrow[\text{ARNG,s}]{} \omega\right) < r_{p>0}\left(\omega \xrightarrow[\text{ARNG,s}]{} \rho\right)^{-1}. \tag{13}$$

Together with our achievability result in Theorem 1, this provides a complete characterisation of the landscape of reversibility in the probabilistic setting: general reversibility is indeed achievable with the asymptotically resource–non-generating transformations $\mathbb{O}_{\text{ARNG}}$, and not only is it impossible under operations that are not allowed to create any resources, but even a slightly more restrictive choice of operations obtained by the change of the underlying resource measure from $R_{\mathbb{F}}^g$ to $R_{\mathbb{F}}^s$ precludes reversibility in general quantum resource theories.

A detailed derivation of Theorem 2, together with technical details and extensions, can be found in Section IV of the Supplementary Information.

**Entanglement distillation**

Two of the most important problems in the understanding of asymptotic entanglement manipulation concern the tasks of extracting 'entanglement bits' (ebits), i.e. copies of the maximally entangled two-qubit singlet state $\Phi_+$, and the reverse task of converting ebits into general noisy states. The rates of these two tasks are known as, respectively, the distillable entanglement $E_{d,\mathbb{O}}^x(\rho) := r_x(\rho \to \Phi_+)$ and the entanglement cost $E_{c,\mathbb{O}}^x(\rho) := r_x(\Phi_+ \to \rho)^{-1}$, where $x$ stands for either $p = 1, p > 0$, or the strong converse rate $p = 1, \dagger$. Although exact expressions can be obtained for the entanglement cost in various settings[10,32,33], the understanding of distillable entanglement appears to be an extremely difficult problem that has so far resisted most attempts at a conclusive solution, except in some special cases[28,34]. Of note is the conjectured result that[10,17]

$$E_{d,\text{NE}}^{p=1}(\rho) \stackrel{?}{=} D_{\text{SEP}}^\infty(\rho), \tag{14}$$

where NE stands for the class of non-entangling operations (equivalent to RNG maps in this theory) and $D_{\text{SEP}}^\infty$ is the regularised relative entropy of entanglement. Establishing this result would recover the deterministic reversibility of entanglement theory (that is, Eq. (5))[10,17]. We note that distillation rates under NE operations are equal to rates under asymptotically non-entangling operations (ANE)[14], which correspond to $\mathbb{O}_{\text{ARNG}}$ in the notation of this work.

We now introduce a close relation that connects entanglement distillation transformations in the probabilistic and strong converse regimes. Namely, we show that one can always improve on the transformation error of a distillation protocol by sacrificing some success probability, and vice versa. What this means in particular is that every rate that can be achieved in the deterministic strong converse regime (i.e. with a possibly large error $\varepsilon < 1$) can also be achieved probabilistically with error going to zero. Crucially, to construct the new, modified protocol from the original one we only need to employ local operations and classical communication (LOCC), which are the standard class of free operations in entanglement theory, meaning that the result applies to essentially all different types of operations that extend LOCC.

**Theorem 3.** Let $\mathbb{O}$ be any class of operations which is closed under composition with LOCC, i.e. such that $\mathcal{E} \in \mathbb{O}, \mathcal{F} \in \text{LOCC} \Rightarrow \mathcal{F} \circ \mathcal{E} \in \mathbb{O}$. This includes in particular the set LOCC itself. Then, for all states $\rho$,

$$E_{d,\mathbb{O}}^{p=1,\dagger}(\rho) = E_{d,\mathbb{O}}^{p>0}(\rho). \tag{15}$$

For the case of (asymptotically) non-entangling operations, we have that

$$E_{d,(\text{A})\text{NE}}^{p=1,\dagger}(\rho) = E_{d,(\text{A})\text{NE}}^{p>0}(\rho) = D_{\text{SEP}}^\infty(\rho). \tag{16}$$

**Proof sketch.** Assume that two spatially separated parties, conventionally called Alice and Bob, share $n$ copies of an entangled state $\rho_{AB}$. Consider any sequence of protocols which allows them to distill entanglement from such states at a rate $r$ with with error $\varepsilon_n \xrightarrow{n\to\infty} \varepsilon$ and probability $p_n \xrightarrow{n\to\infty} p$ (this includes the deterministic strong converse case where $p_n = 1$).

After performing the considered distillation protocol, they share a many-copy state $\tau_{A'B'}$ which approximates $\Phi_+^{\otimes \lfloor rn \rfloor}$. What they can do now is to sacrifice a fixed number $k$ of their qubit pairs in order to perform a state discrimination protocol: by testing whether the $k$-copy subsystem is in the state $\Phi_+^{\otimes k}$ and discarding their whole state when it is not, they can probabilistically bring the state of the rest of their shared system closer to $\Phi_+^{\otimes \lfloor rn \rfloor - k}$. We show that this can be done by a simple LOCC protocol wherein Alice and Bob perform measurements in the computational basis and compare their outcomes. Since $k$ is arbitrary here, Alice and Bob can perform the modified protocol without reducing the asymptotic transformation rate. Conversely, in a similar manner they can also increase their probability of success by sacrificing some transformation fidelity. By deriving the exact conditions for when distillation protocols can be refashioned in such a away, we observe that another sequence of entanglement distillation protocols with error $\varepsilon_n' \xrightarrow{n\to\infty} \varepsilon'$ and probability $p_n' \xrightarrow{n\to\infty} p'$ can exist if and only if

$$p(1-\varepsilon) = p'(1-\varepsilon'). \tag{17}$$

This directly implies Eq. (15).

To see Eq. (16), it suffices to combine Theorem 1 with the known result that $D_{\text{SEP}}^\infty(\rho)$ is a strong converse rate for distillation under ANE[10,35].

We have already shown the probabilistic reversibility of entanglement theory in Theorem 1, so let us now discuss the deterministic case. Here, reversibility is fully equivalent to the question of whether $E_{d,\mathbb{O}}^{p=1}(\rho) = E_{c,\mathbb{O}}^{p=1}(\rho)$ holds for all quantum states, which has been conjectured to be true for the class of asymptotically non-entangling operations[10]. Combining our results with the known findings of Brandão and Plenio[10], we have that

$$E_{d,\text{ANE}}^{p>0}(\rho) \stackrel{\text{Thm. 3}}{=} E_{d,\text{ANE}}^{p=1,\dagger}(\rho) \stackrel{[10,17]}{=} E_{c,\text{ANE}}^{p=1}(\rho)$$
$$\stackrel{[10]}{=} D_{\text{SEP}}^\infty(\rho) \stackrel{\text{Thm. 1}}{=} E_{c,ANE}^{p>0}(\rho). \tag{18}$$

The missing link is thus the question if $E_{d,\text{ANE}}^{p=1}(\rho) \stackrel{?}{=} E_{d,\text{ANE}}^{p>0}(\rho)$, or the equivalent[17] question of whether $E_{c,ANE}^{p=1,\dagger}(\rho) \stackrel{?}{=} E_{c,ANE}^{p>0}(\rho)$. Showing either of these statements would complete the proof of the deterministic reversibility of quantum entanglement under asymptotically non-entangling operations. An interesting consequence of the above is that establishing the equivalent of Theorem 3 for entanglement dilution would be sufficient to recover a fully reversible entanglement theory.

We remark that other quantum resource theories may not be amenable to a characterisation in terms of distillation and dilution because they may not possess a suitably well-behaved unit of a resource resembling the maximally entangled state[16,36,37]. Nonetheless, reversibility in all resource theories can be understood as in our Theorem 1.

## Discussion

We have shown that the conjectured reversibility of general quantum resources can be recovered, albeit in a probabilistic manner that employs probabilistic protocols with non-vanishing probability of success. This allowed us to identify the setting of probabilistic resource transformations as one that is completely governed by a unique entropic quantity − the regularised relative entropy − thus solidifying the parallels between thermodynamics and diverse types of quantum resources. We further showed that the choice of asymptotically resource–non-generating operations is optimal in this setting, in the sense that all smaller classes of probabilistic operations are necessarily irreversible, providing a strong no-go restriction on how reversibility could be achieved.

Although this precise characterisation of asymptotic rates is appealing, our setting departs from the original reversibility conjectures of refs. 10,16, since it considers transformations that are only required to be achieved with some probability. This may not be enough to ensure the existence of a repeatable, reversible transformation cycle in practice. Nevertheless, in view of the close relations between probabilistic and deterministic rates (Eq. (2)) we regard our results as evidence that reversibility could indeed be recovered also in the deterministic setting. This is further motivated by the fact that, in many quantum information processing tasks, strong converse rates actually coincide with the optimal achievable rates[38–45], meaning that $r_{p=1} = r_{p=1}^{\dagger}$ and the hierarchy in Eq. (2) collapses. However, a complete proof of this fact in the setting of resource manipulation remains elusive, and it is still possible that one of the inequalities in Eq. (2) may be strict for some states, thus ruling out deterministic reversibility. We hope that our results stimulate further research in this direction, leading to an eventual resolution of the open questions that cloud the understanding of asymptotic resource manipulation[18].

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

## Acknowledgements
We are grateful to the anonymous referees of the conference Quantum Information Processing 2024 for helpful comments. We acknowledge discussions with Tulja Varun Kondra and Alexander Streltsov. We further thank the Freie Universität Berlin for hospitality.

## Author contributions
Both authors contributed to all aspects of this manuscript. The first idea for the project was sketched out in a joint discussion by L.L. and B.R. B.R. later developed the idea, derived the main result, and wrote the first draft of the manuscript.

## Competing interests
The authors declare no competing interests.
