## [Peer Review File · Nature Communications]

Reversibility of quantum resources through probabilistic protocolsREVIEWER COMMENTS

Reviewer #1 (Remarks to the Author):

A central question in the field of quantum resource theories is the reversibility of transformations between resourceful states and the existence of resource quantifiers that govern the corresponding transformation rates. In this manuscript the authors make significant progress in this direction by establishing the reversibility of general quantum resource theories (under relatively mild assumptions) and show that the regularised relative entropy to completely determine transformation rates in general quantum resource theories. This is achieved by considering probabilistic transformations rather than deterministic ones. To prevent transformations from becoming unphysical the authors consider transformation rates with probabilities strictly bounded from zero. Theorem 1 then states that the transformation rates are completely determined by the regularised relative entropies of the initial and target states, which implies general reversibility. Moreover, the work is build on considering asymptotically resource non-generating operations, where one allows a small amount of resources to be generated, measured by the generalised robustness, which must vanish in the limit n to infinity. The authors show that their choice of setting is (essentially) optimal, by showing that choosing smaller sets of operations, by replacing the generalised robustness by the standard robustness, would not lead to general reversibility (Theorem 2). Finally, the results are discussed in the context of entanglement distillation.

The results of this manuscript are quite impressive and the manuscript itself is really enjoyable to read. The paper makes significant contributions to the understanding of general quantum resource theories and their reversibility. All the claims are well supported and proofs seem correct (at least to the extent that I have checked). Therefore, I am very happy to recommend acceptance of this article.

Reviewer #2 (Remarks to the Author):

I had the pleasure of reviewing this article for the QIP 2024 conference (which was ultimately accepted as a contributed talk). I will essentially paste that review below since my

assessment of the article remains the same. In short, I think this is a very nice scientific work meeting the standards for publication in Nature Communications. However, my concern over the "reversibility" interpretation of the result and the physical motivation for the considered operations are still withstanding.

Before giving my full recommendation for publication, I would like the authors to respond to the two weaknesses listed below.

This work can arguably be viewed as Part Three in a series of papers involving the authors on the topic of asymptotic resource reversibility [1, 2]. Note that the first two were accepted as QIP talks in the past two years. In the first work [1], the authors showed that entanglement is irreversible under non-entangling operations. The second paper [2] exposes a technical mistake in the proof of Ref. [3], which leaves it unresolved whether general reversibility holds under asymptotically resource non-generating (ARNG) operations. This current paper show that if we consider probabilistic ARNG operations, then the main conclusion of [3] can be restored and asymptotic reversibility holds.

Strengths:

- This paper establishes its main result by carefully showing how noisy asymptotic transformations can be converted into probabilistic ones with less error. In [3] it was correctly proven that the regularized relative entropy (REE) quantifies the strong converse rate for state convertibility under ARNG. Moreover, an explicit construction was given in [3] showing that the strong converse rate is "achievable" in the sense that the error in the target state is bounded strictly away from one. The tricky (and unresolved) part is showing that this error rate goes to zero (if it even does). Instead of solving that problem directly, this paper takes a different approach and converts the construction of [3] into a probabilistic (i.e. flagged or post-selected) map with zero error. The technique employed is quite clever. Using the standard hypothesis testing maps, the authors introduce a modulation factor μ_n to the failure term. This causes the map to be non-trace-

preserving, but it also amplifies the relative contribution of the target state. The authors prove that the μ_n can be modulated in such a way that $\mu_n \rightarrow 0$ while the resulting sequence of maps is still ARNG.

- This exchange between transformation error rate and post-selection probability is proven more generally for entanglement distillation using any set of operations that includes LOCC (Theorem 2). I believe this result is of value/interest independent of the reversibility problem.

- In general, the paper is clearly written with a thorough background and thoughtful discussion. It does an admirable job of presenting a complex technical problem in a fairly readable way, and I would expect a QIP talk to be likewise accessible.

Weaknesses:

- Putting aside my technical praise, I'm not exactly sure how to judge the significance of the main result. I would challenge the author's conclusion that they've shown reversibility in any standard sense of "reversibility". In practice, implementing these probabilistic maps would involve an experimenter throwing away the output some (most?) of the time. But since these processes are lossy, how can they be deemed reversible? In particular, the failure probability of $\rho \rightarrow \omega$ can be different than the failure probability of the reverse transformation $\omega \rightarrow \rho$. While the probabilistic maps in the achievability proof have a non-zero probability of success, it seems there is no universal lower bound (for a fixed dimension), and so these probabilities could be substantially different.

An analog here would be exchanging USD to EUROS at an exchange rate r , but then having to pay a (potentially large) fractional service charge. I have trouble understanding how processes like this are reversible in any resource-theoretic way.

- I am also unsure how to physically justify the operations studied in this work. We're considering ARNG operations, which I believe only have physical motivation when considered from a resource theory perspective. But as expressed in my previous comment,

it's not clear how to place this work in a resource-theoretic context given that we're allowed to throw things away with any probability short of one. It would be great to hear the authors comment on this. Also, it should be noted that the ARNG operations studied here are defined w.r.t. the global robustness of resource. This is consistent with the approach taken in [3]. However, as the authors have convincingly argued in [2], this is problematic since such operations allow for large "macroscopic" fluctuations of resource production. A more natural restriction is to define ARNG w.r.t. resource robustness. But in this case it appears the proof technique used here won't work. Do we still have probabilistic reversibility? Or can the no-go construction of [2] also be applied in this case?

[1] doi:10.1038/s41567-022-01873-9

[2] doi:10.22331/q-2023-09-07-1103

[3] doi:10.1007/s00220-010-1005-z

Reviewer #3 (Remarks to the Author):

The problem motivating this work, one of reversibility in general resource theories, is an interesting one. There is now a wide audience of researchers interested in insights from this line of work. The authors claim to recover reversibility in some way. They do a great job of providing details and clearly stating their approach and results obtained.

My chief critique is about the interpretation of results.

The way the authors are (implicitly?) calculating rates for conversion is at odds with standard practice in the field (for instance see 'yield' in Ref. 17) . As the authors may be well aware, in the present context, the standard definition of rate is the fraction of initial states that get converted into a final state with asymptotically vanishing error given free access to some set of resources. If there is a protocol for doing such a conversion which may fail, but when it succeeds say with probability 'p', it does so with some non-zero rate 'a', then the rate of the full protocol is $p \times a$. In this work this probabilistic factor is dropped when discussing rates. Since rates in this work don't correspond to the fraction of states converted I am

unable to make sense of statements based on these rates, chiefly the statement of reversibility. It is hard to understand statements such as

'reversibly inter convert all states in general quantum resource theories, as long as one allows protocols that may only succeed probabilistically'

in the abstract,

'we resolve the question by constructing the first complete reversible framework for general quantum resources, including entanglement'

in the introduction, and so on.

Next, I gather that various remarkable aspects of achieving reversible transformations (such as identification of a unique entropic measure) only come to fruition when the definition of rate is along the lines illustrated above. Perhaps the results in this work are valuable in other ways. For instance do they rule out bound entanglement type phenomenon (ref. 10,11) for the class of free resources under consideration?

This critique about the interpretation of the results shouldn't reduce the technical achievements claimed in this work. The problem motivating this work is now recognized to be tougher than expected. If correct, as the authors state, this work provides positive evidence in favor of reversibility of general quantum resource theories.

RESPONSE TO REVIEWER #1

A central question in the field of quantum resource theories is the reversibility of transformations between resourceful states and the existence of resource quantifiers that govern the corresponding transformation rates. In this manuscript the authors make significant progress in this direction by establishing the reversibility of general quantum resource theories (under relatively mild assumptions) and show that the regularised relative entropy to completely determine transformation rates in general quantum resource theories. This is achieved by considering probabilistic transformations rather than deterministic ones. To prevent transformations from becoming unphysical the authors consider transformation rates with probabilities strictly bounded from zero. Theorem 1 then states that the transformation rates are completely determined by the regularised relative entropies of the initial and target states, which implies general reversibility. Moreover, the work is build on considering asymptotically resource non-generating operations, where one allows a small amount of resources to be generated, measured by the generalised robustness, which must vanish in the limit n to infinity. The authors show that their choice of setting is (essentially) optimal, by showing that choosing smaller sets of operations, by replacing the generalised robustness by the standard robustness, would not lead to general reversibility (Theorem 2). Finally, the results are discussed in the context of entanglement distillation.

The results of this manuscript are quite impressive and the manuscript itself is really enjoyable to read. The paper makes significant contributions to the understanding of general quantum resource theories and their reversibility. All the claims are well supported and proofs seem correct (at least to the extent that I have checked). Therefore, I am very happy to recommend acceptance of this article.

We thank the Reviewer for the kind words about our paper and for the positive evaluation of our results.

RESPONSE TO REVIEWER #2

I had the pleasure of reviewing this article for the QIP 2024 conference (which was ultimately accepted as a contributed talk). I will essentially paste that review below since my assessment of the article remains the same. In short, I think this is a very nice scientific work meeting the standards for publication in Nature Communications. However, my concern over the "reversibility" interpretation of the result and the physical motivation for the considered operations are still withstanding.

Before giving my full recommendation for publication, I would like the authors to respond to the two weaknesses listed below.

We are grateful for the Reviewer's careful evaluation of our work, both now and in the QIP review process. We already tried to address some of the reservations in the interpretation of our results after the QIP reviews. However, we understand that this was insufficient and we aim to rectify it with this resubmission.

Strengths:

- This paper establishes its main result by carefully showing how noisy asymptotic transformations can be converted into probabilistic ones with less error. In [3] it was correctly proven that the regularized relative entropy (REE) quantifies the strong converse rate for state convertibility under ARNG. Moreover, an explicit construction was given in [3] showing that the strong converse rate is "achievable" in the sense that the error in the target state is bounded strictly away from one. The tricky (and unresolved) part is showing that this error rate goes to zero (if it even does). Instead of solving that problem directly, this paper takes a different approach and converts the construction of [3] into a probabilistic (i.e. flagged or post-selected) map with zero error. The technique employed is quite clever. Using the standard hypothesis testing maps, the authors introduce a modulation factor μ_n to the failure term. This causes the map to be non-trace-preserving, but it also amplifies the relative contribution of the target state. The authors prove that the μ_n can be modulated in such a way that $\mu_n \rightarrow 0$ while the resulting sequence of maps is still ARNG.
- This exchange between transformation error rate and post-selection probability is proven more generally for entanglement distillation using any set of operations that includes LOCC (Theorem 2). I believe this result is of value/interest independent of the reversibility problem.
- In general, the paper is clearly written with a thorough background and thoughtful discussion. It does an admirable job of presenting a complex technical problem in a fairly readable way, and I would expect a QIP talk to be likewise accessible.

We appreciate the kind words about our results and our manuscript.

Weaknesses:

- Putting aside my technical praise, I'm not exactly sure how to judge the significance of the main result. I would challenge the author's conclusion that they've shown reversibility in any standard sense of "reversibility". In practice, implementing these probabilistic maps would involve an experimenter throwing away the output some (most?) of the time. But since these processes are lossy, how can they be deemed reversible? In particular, the failure probability of $\rho \rightarrow \omega$ can be different than the failure probability of the reverse transformation $\omega \rightarrow \rho$. While the probabilistic maps in the achievability proof have a non-zero probability of success, it seems there is no universal lower bound (for a fixed dimension), and so these probabilities could be substantially different.

An analog here would be exchanging USD to EUROS at an exchange rate r , but then having to pay a (potentially large) fractional service charge. I have trouble understanding how processes like this are reversible in any resource-theoretic way.

We agree with the Reviewer that our state conversion protocols are by their nature intrinsically lossy, and that the success probabilities, albeit asymptotically constant, are not explicitly controlled in our definition of 'rate'. This means that, indeed, ours is not a rate in a traditional sense. However, there is a sense in which our rate is physically very meaningful, for it coincides with the number of input states that have to be *simultaneously* coherently manipulated in order to obtain a prescribed

number of output states, in a setting where a constant overhead on the number of copies of the input state does not entail a huge cost. We expounded this interpretation of the probabilistic rate in the revised version of the main text (see also the response to Reviewer #3 below).

The above scenario is relevant in a setting where the cost of coherent manipulation far exceeds that of input state production, and there is no effective quantum memory to store the input nor the output, which has to be produced all at once. The analogy with the currency exchange rates the Reviewer proposes does not reflect these aspects of the problem, which may however be very relevant when constructing quantum hardware to manipulate quantum resources. Although different than conventional definitions, we show our definition to be well-behaved and closely connected to standard deterministic (weak and strong converse) rates — see in particular Eq. (2) of the paper — which further motivates our probabilistic rates as a sensible choice.

Even more importantly, despite the different definitions, we believe that one of the strongest contributions of our results is that they serve as **evidence that reversibility is an achievable phenomenon** in the theory of quantum entanglement. This is because here we construct the **very first reversible framework** whatsoever — before this work, it was not known if any form of a reversible framework exists, no matter how much one relaxes the setting and the definitions. As the original reversibility conjecture of Brandão–Plenio has now been understood to be much more difficult than originally thought, there is thus a natural motivation to investigate reversibility in slightly relaxed settings: Is there any way to obtain reversible entanglement transformations at all, or is the theory simply completely irreversible? What does it take to achieve reversibility?

Our framework considers an approach which relaxes the definition of asymptotic rates but, importantly, does not relax any other assumptions — all of the operational constraints on entanglement manipulation in our work are the same as considered by Brandão and Plenio. Our approach is thus not more permissive from a strictly resource-theoretic point of view. And the evidence provided by our result is not simply conceptual: because of the close quantitative connections between our rates and deterministic ones, we consider our results to be directly indicating that reversibility also in the strongest, deterministic sense should be within reach.

Ultimately, our main goal was not to show that reversibility ‘in a standard sense’ (as the Reviewer put it) is possible, but rather that there exists an alternative — but still reasonable — setting in which asymptotic transformations of quantum states can be governed by a single, unique entropic resource monotone. Our definition of rates is then just a tool used to establish this particular setting where this is possible. At no point do we claim to recover reversibility as originally conjectured, and indeed we tried to clarify and emphasise our main assumptions and motivations in all parts of the manuscript. What we do find, however, is a reversible framework that is the first of its kind and gives some hope in the pursuit of a complete theory of reversibility in quantum resource manipulation. This, we concur with the Reviewer, should be the ultimate aim of this approach.

To address these points in the manuscript, we have further expanded the discussion of the definition of probabilistic rates (Introduction and section ‘Resource transformation rates’) as well as the consequences for reversibility (Introduction and Discussions).

We hope that this addresses the Reviewer’s reservations and that we now provide a sufficient motivation for and interpretation of our setting.

- I am also unsure how to physically justify the operations studied in this work. We’re considering ARNG operations, which I believe only have physical motivation when considered from a resource theory perspective. But as expressed in my previous comment, it’s not clear how to place this work in a resource-theoretic context given that we’re allowed to throw things away with any probability short of one. It would be great to hear the authors comment on this. Also, it should be noted that the ARNG operations studied here are defined w.r.t. the global robustness of resource. This is consistent with the approach taken in [3]. However, as the authors have convincingly argued in [2], this is problematic since such operations allow for large “macroscopic” fluctuations of resource production. A more natural restriction is to define ARNG w.r.t. resource robustness. But in this case it appears the proof technique used here won’t work. Do we still have probabilistic reversibility? Or can the no-go construction of [2] also be applied in this case?

Thank you for raising this important point, which we already addressed after the QIP review by the Reviewer: in our initial Nature Communications submission, we had added a new technical section (part IV of the supplementary information) and a new subsection in the main text ('Optimality of ARNG transformations'). In these additions, we generalised our previous results from [Nat. Phys. 19, 184–189 (2023)] to show that one gets *irreversibility* for more restrictive definitions of ARNG operations. This means that the choice of ARNG w.r.t. the generalised robustness (as used in our paper) is essentially optimal. We believe that this directly answers the Reviewer's question. Although we would have loved to study a more 'grounded' and operationally (rather than axiomatically) motivated class of operations, such a possibility is ruled out.

We thank the Reviewer for this suggestion, as we believe that it has significantly strengthened the results of the paper.

RESPONSE TO REVIEWER #3

The problem motivating this work, one of reversibility in general resource theories, is an interesting one. There is now a wide audience of researchers interested in insights from this line of work. The authors claim to recover reversibility in some way. They do a great job of providing details and clearly stating their approach and results obtained.

My chief critique is about the interpretation of results. The way the authors are (implicitly?) calculating rates for conversion is at odds with standard practice in the field (for instance see 'yield' in Ref. 17). As the authors may be well aware, in the present context, the standard definition of rate is the fraction of initial states that get converted into a final state with asymptotically vanishing error given free access to some set of resources. If there is a protocol for doing such a conversion which may fail, but when it succeeds say with probability 'p', it does so with some non-zero rate 'a', then the rate of the full protocol is $p \times a$. In this work this probabilistic factor is dropped when discussing rates. Since rates in this work don't correspond to the fraction of states converted I am unable to make sense of statements based on these rates, chiefly the statement of reversibility. It is hard to understand statements such as

'reversibly inter convert all states in general quantum resource theories, as long as one allows protocols that may only succeed probabilistically'

in the abstract,

'we resolve the question by constructing the first complete reversible framework for general quantum resources, including entanglement'

in the introduction, and so on.

Next, I gather that various remarkable aspects of achieving reversible transformations (such as identification of a unique entropic measure) only come to fruition when the definition of rate is along the lines illustrated above. Perhaps the results in this work are valuable in other ways. For instance do they rule out bound entanglement type phenomenon (ref. 10,11) for the class of free resources under consideration?

This critique about the interpretation of the results shouldn't reduce the technical achievements claimed in this work. The problem motivating this work is now recognized to be tougher than expected. If correct, as the authors state, this work provides positive evidence in favor of reversibility of general quantum resource theories.

We would like to begin by stressing that providing 'positive evidence in favour of reversibility of general quantum resource theories', as the Reviewer put it, is precisely what we believe is one of the most significant contributions of our work. We do not claim to resolve the original question of reversible resource transformation, but rather provide a slightly modified framework that achieves the desired aim of obtaining a unique entropic measure of quantum resources. We strongly believe that this in itself is an important and novel development that helps us understand fundamental features of quantum entanglement and other resources.

However, we understand the reservation of the Reviewer regarding the probabilistic rates. To address this, let us first provide some intuition for why we believe that our definition of an asymptotic rate, although different from conventionally used ones, is not an unreasonable one.

First, let us look at how a standard, deterministic rate $r(\rho \rightarrow \omega)$ is defined. The common intuitive definition is that it is the least number r such that, per each copy of ρ , we can asymptotically obtain rn copies of ω ; equivalently, we need r^{-1} copies of ρ per each copy of ω . More precisely, there must exist a sequence of transformations $(\mathcal{E}_n)_n$ such that $\mathcal{E}_n(\rho^{\otimes n}) \approx \omega^{\otimes rn}$. It is important to remember that in this setting, the two following interpretations of the (inverse) rate $r(\rho \rightarrow \omega)^{-1}$ are equivalent:

- it tells us how many copies of ρ we need to generate per each copy of ω (in an asymptotic sense)
- it tells us how many copies of ρ we need to *manipulate* per each copy of ω (in an asymptotic sense).

Here, 'manipulate' means to implement a quantum operation which can coherently transform n copies of ρ as $\mathcal{E}_n(\rho^{\otimes n})$. Because of this, the operational cost of the protocol can be understood

in two different, but fully equivalent ways: either as the number of copies of ρ that need to be generated, or as the size of the quantum operations that have to be implemented.

However, this drastically changes when probabilistic protocols are considered. Here, the two interpretations discussed above give rise to two *inequivalent* notions of a rate: we can either ask about how many copies we need to generate, or how many copies we need to manipulate. To see that these notions are not the same any more, consider, for instance, two different protocols: (1) a probabilistic protocol which transforms n copies of ρ to rn copies of ω with probability $1/2$, and (2) a deterministic protocol which transforms $2n$ copies of ρ to rn copies of ω . They use the same number of copies of ρ , as the probabilistic protocol needs to be repeated twice on average. However, we hope that the Reviewer will agree with us that the second (deterministic) protocol is more difficult to implement in practice than the first one, since it requires the coherent manipulation of $2n$ copies of a state rather than only n copies. Does it then make sense to evaluate the rates of the protocols on equal footing? We do not believe so, since the rate is supposed to quantify the difficulty in performing the given state transformation. Especially when working under the assumption that one is able to generate asymptotically large ($n \rightarrow \infty$) numbers of copies of quantum states, the number of copies itself may not be the most suitable figure of merit.

This leads us to the conclusion that the ‘rate’ — in the sense of ‘cost per copy’ — of a probabilistic protocol should focus on how many copies of a state we need to manipulate. But of course it is not fair to completely disregard the probability of success, as this would allow for unphysically low probability of success (e.g. going to 0 exponentially fast). What we do, then, is to say that a probabilistic protocol has rate r if two conditions are satisfied:

- (i) The protocol manipulates only n copies of ρ per each rn copies of ω ; precisely, $\frac{\mathcal{E}_n(\rho^{\otimes n})}{\text{Tr} \mathcal{E}_n(\rho^{\otimes n})} \approx \omega^{\otimes rn}$.
- (ii) The protocol succeeds with an overall probability larger than some non-zero constant; precisely, $\liminf_{n \rightarrow \infty} \text{Tr} \mathcal{E}_n(\rho^{\otimes n}) > 0$.

Asymptotically, when $n \rightarrow \infty$, the fact that the protocol has to be repeated a constant number of times is insignificant compared to the difficulty in implementing the many-copy transformations $\mathcal{E}_n(\rho^{\otimes n})$, so we choose to discount the probability of success and focus on the number of manipulated copies.

This is the definition that we adopt in our work and for which all of our results are stated. We believe that it is an intuitive definition of a rate in this setting and that it does not go beyond reasonable notions of asymptotic yield.

However, we do acknowledge that our definition is different from ones that were typically studied in the literature, especially since the latter definitions typically focus on deterministic protocols. We have already tried to clarify this in the manuscript: we stressed that our results establish an *alternative* reversibility framework, and for it to be valid one needs to allow for probabilistic protocols with this specific definition of a rate. We do not believe that this diminishes the result, since our crucial point is that **there were no reversible frameworks whatsoever before ours** — even under relaxed constraints or definitions, there has been no setting in which entanglement could be made reversible. This cast significant doubt on the previously conjectured reversible framework of [Brandão and Plenio, Nat. Phys. 4, 873 (2008)] and on the connection between entanglement theory and thermodynamics. We now show that such a connection *is* possible: we construct a setting in which all asymptotic transformations are governed by a single, unique entropic measure of entanglement.

Although our approach may require an alternative way of computing asymptotic rates, it is important to note that it does not relax any other assumptions: all of the operational constraints on entanglement manipulation in our work are the same as considered by Brandão and Plenio. Our approach is thus not a more permissive framework from a resource-theoretic point of view.

It should also be noted that one of our technical results is to show that the probabilistic rates, as defined in our way, fit naturally within the information-theoretic framework of transformation rates: they are closely connected to and bounded by the standard deterministic (weak and strong converse) rates — see e.g. Eq. (2) of the paper. We believe that this further motivates our

definition of rates and strengthens the conceptual significance of our reversibility results, as it directly suggests that our rates are not much different from conventional ones, and hence our notion of reversibility does not stray far from the more conventional reversibility.

In summary, our definitions should be regarded simply as a technical tool that we use to establish the feasibility of a reversible framework of quantum resources; however, they are not conceptually different or much more permissive than conventional definitions. As the Reviewer points out, the original problem of determining whether a deterministic reversible framework of entanglement exists seems to be a very tough one. This provides a strong motivation for looking at slightly more relaxed frameworks of entanglement manipulation, which could nevertheless shed some light on the problem of reversibility in entanglement theory. Here we establish precisely that — a complete, rigorous framework which shows that a sensible notion of reversibility is indeed achievable, and one that provides evidence for why the original conjecture should also be true. The only ‘price’ one has to pay in our framework is to accept our modified definition of a rate. But we hope to have justified why that is not a far-fetched concession to make.

In order to clarify this point in the manuscript and avoid any potential misunderstandings, we have provided further clarifications in the manuscript, in particular surrounding our discussion of the definition of the probabilistic rates (Introduction and section ‘Resource transformation rates’) as well as the consequences for reversibility (Introduction and Discussions).

We hope that this provides a sufficient justification for our choice of the definition, and it helps clarify the significance of our results.

For instance do they rule out bound entanglement type phenomenon (ref. 10,11) for the class of free resources under consideration?

Let us clarify here that, even in the standard (deterministic) setting, it is already known that bound entanglement does not exist under non-entangling operations (see ref. 15 or ref. 31). However, even though we know that the distillation rate is non-zero in that case, we do not how to actually compute the rate — the exact value of deterministic distillable entanglement is not known, because computing it is equivalent to solving the notorious generalised quantum Stein’s lemma.

By going to our ‘probabilistic’ regime, we simplify the evaluation of the rate and we show that the distillable entanglement of any state is given exactly by the regularised relative entropy of entanglement D_{SEP}^{∞} . This does require accepting the slightly different definition of a rate, but in return in provides a simpler, precise understanding of the asymptotic power in entanglement manipulation.

REVIEWERS' COMMENTS

Reviewer #2 (Remarks to the Author):

I greatly appreciate the detailed responses from the authors, and I believe their rebuttal is convincing. Overall, I believe this manuscript represents important progress in the asymptotic theory of entanglement. The non-standard notion of reversibility studied in this paper makes the result more specialized than if the full reversibility question were resolved. But I still believe it's above the standards of Nat. Comm, and I therefore recommend it for publication.

One request: Given that the term "reversibility" has a fairly universal meaning in the information-theoretic sense (and this is not the problem solved in this paper), I would suggest the authors change their title to be "***Reversing** quantum resources through probabilistic protocols," or something similar.

Reviewer #3 (Remarks to the Author):

The revised version merits publication. The additional discussion of rates in this version is an improvement. It may not be surprising if the types of alternative routes to the Stein's Lemma taken in this work become subjects of further study.

RESPONSE TO REVIEWER #2

I greatly appreciate the detailed responses from the authors, and I believe their rebuttal is convincing. Overall, I believe this manuscript represents important progress in the asymptotic theory of entanglement. The non-standard notion of reversibility studied in this paper makes the result more specialized than if the full reversibility question were resolved. But I still believe it's above the standards of Nat. Comm, and I therefore recommend it for publication.

We appreciate the Reviewer's evaluation of the rebuttal and the positive judgement of our results.

One request: Given that the term "reversibility" has a fairly universal meaning in the information-theoretic sense (and this is not the problem solved in this paper), I would suggest the authors change their title to be "Reversing quantum resources through probabilistic protocols", or something similar.

We have carefully considered this request, but we do not find that this change would meaningfully disambiguate the issue — if a reader has some precise notion of 'reversibility' in mind, then surely they would get equally confused by the word 'reversing'. We do agree that it is very important to qualify the exact type of reversibility that we solve in the paper, which is why we always stress the need for probabilistic operations and rates, and why we emphasise (in the Introduction, Results, and Discussions sections) that we do not exactly recover the stronger form of reversibility conjectured before.

We believe that including '*through probabilistic protocols*' in the title is sufficient to alert the reader to the fact that this is a specific type of reversibility, and we have thus decided to keep the title unchanged. We nevertheless appreciate the suggestion, which has prompted us to strengthen some wording in the manuscript for the sake of clarity.

RESPONSE TO REVIEWER #3

I The revised version merits publication. The additional discussion of rates in this version is an improvement. It may not be surprising if the types of alternative routes to the Stein's Lemma taken in this work become subjects of further study.

We are happy that our resubmission has addressed the Reviewer's reservations. We thank the Reviewer for their evaluation of our work.